# Therapeutic effects of PDGF-AB/BB against cellular senescence in human intervertebral disc

**Changli Zhang[1,2], Martha Elena Diaz-Hernandez[1,2], Takanori Fukunaga[1,2], Sreekala Shenoy[1,2], Sangwook Tim Yoon[3], Lisbet Haglund[4], Hicham Drissi[1,2]\***

[1]Department of Orthopaedics, Emory University School of Medicine, Atlanta, United States; [2]Atlanta VA Medical Center, Decatur, United States; [3]Emory Orthopaedics and Spine Center, Atlanta, United States; [4]Department of Surgery, McGill University, Montreal, Canada

## eLife Assessment

This work demonstrates the therapeutic potential of recombinant human PDGF-AB/BB proteins in alleviating the senescent signatures of primary human intervertebral disc cells. The study represents a **fundamental**, significant advance in the treatment of intervertebral disc degeneration through the suppression of senescence. The strength of evidence supporting these conclusions is **compelling**, as it is primarily based on transcriptomic analysis and direct protein measurements from relatively homogeneous cell populations. This work will be of interest to spine basic scientists and clinicians, as well as to musculoskeletal scientists more broadly. The revised manuscript adds greater clarity, and the impact of the study is greatly enhanced.

**\*For correspondence:**
hicham.drissi@emory.edu

**Competing interest:** The authors declare that no competing interests exist.

**Abstract** Accumulation of senescent cells is closely linked with intervertebral disc (IVD) degeneration, a prevalent age-dependent chronic disorder causing low back pain. While previous studies have highlighted that platelet-derived growth factor (PDGF) mitigated IVD degeneration through anti-apoptotic and pro-anabolic effects, its impact on IVD cell senescence remains elusive. In this study, human NP and AF cells derived from aged, degenerated IVDs were treated with recombinant human (rh) PDGF-AB/BB for 5 d. Transcriptome profiling by mRNA sequencing revealed that NP and AF cells responded to the treatment in similar yet distinct ways. The effects of PDGF-AB and BB on human IVD cells were comparable. Specifically, rhPDGF-AB/BB treatment downregulated genes related to neurogenesis and mechanical stimulus response in AF cells, while in NP cells, metabolic pathways were predominantly suppressed. In both NP and AF cells, rhPDGF-AB/BB treatment upregulated genes involved in cell cycle regulation and response to reduced oxygen levels, while downregulating genes related to senescence-associated phenotype, including oxidative stress, reactive oxygen species (ROS), and mitochondria dysfunction. Network analysis revealed that PDGFRA and IL6 were the top hub genes in treated NP cells. Furthermore, in irradiation-induced senescent NP cells, PDGFRA gene expression was significantly reduced compared to non-irradiated cells. However, rhPDGF-AB/BB treatment increased PDGFRA expression and mitigated the senescence progression through increased cell population in the S phase, reduced SA-β-Gal activity, and decreased expression of senescence-related regulators. Our findings reveal a novel anti-senescence role of PDGF in the IVD, making it a promising potential candidate to delay aging-induced IVD degeneration.

**eLife digest** Back pain is among the most common health problems and is frequently caused by damage to the discs between the vertebrae, known as intervertebral discs. These discs act as cushions, allowing for movement and absorbing mechanical forces.

As people age, these discs become stiff and wear out, which is a major risk factor for back pain. Cellular senescence, a process in which cells stop dividing and start releasing harmful cell products, contributes to this degeneration by damaging the surrounding cells. In aged or degenerated intervertebral discs, the number of senescent cells is significantly increased.

The protein PDGF is a growth factor involved in cell growth and division and is widely used in the clinical setting for tissue regeneration and repair. So far, it was unknown if PDGF could slow down cellular senescence in intervertebral discs.

To find out more, Zhang et al. collected two types of cells found in intervertebral discs – the nucleus pulposus cells and the annulus fibrosus cells – from tissue samples of healthy individuals and patients with disc hernias. They then treated these cells with PDGF and analyzed their gene activity. The experiments showed that PDGF slowed cellular senescence in both cell types. In both tissues, PDGF increased the expression of genes that promote the cell cycle and decreased the expression of genes related to senescence, including those involved in oxidative stress responses and mitochondrial dysfunction. Zhang et al. also noted some tissue-specific responses to PDGF treatment, with nucleus pulposus cells responding more by reducing the activity of genes involved in metabolic pathways, and annulus fibrosus cells by deactivating genes involved in responses to mechanical stimulation experienced during daily activities.

Zhang et al. further developed a 'cellular senescence model' by exposing healthy cells extracted from the nucleus pulposus and the annulus fibrosus to a single dose of senescence-inducing radiation in both cell types. Immediate treatment with PDGF reduced the expression of senescence-related genes.

In summary, the study indicates that PDGF can inhibit cell senescence in intervertebral discs. The next step will be to validate these findings in animal models and clinical trials to ensure safety and effectiveness. In the future, this could lead to new strategies for developing minimally invasive therapies to inhibit or halt the progression of degenerative disc disease within the ageing spine.

## Introduction

Low back pain (LBP), ranked as the first cause of years lived with disability, is a prevalent condition. Although the etiology of LBP is multifactorial, a major contributor of LBP is IVD degeneration, which is increasing exponentially due to a growing aged population worldwide (*Feng et al., 2016*). Current treatments, such as medicine, physical therapy, and surgical excision of the herniated tissues, only temporarily alleviate painful symptoms (*Xin et al., 2022*). In fact, patients often experience persistent LBP after surgery (*Coronado et al., 2015*). The fundamental cause of this is the inadequate knowledge of the underlying pathophysiology mechanisms leading to IVD degeneration. Therefore, it is crucial to understand the pathogenesis of IVD degeneration and develop novel therapeutic interventions to mitigate the degenerative processes.

The IVD consists of three compartments: the gelatinous nucleus pulposus (NP), fibrous annulus fibrosus (AF), and cartilaginous endplate (*Sivan et al., 2014*). NP and AF tissues play distinct yet complementary roles in maintaining the spinal function. The NP has a high content of proteoglycans and water, absorbing and distributing mechanical forces. In contrast, AF consists of multiple layers of concentric lamellae, which provide strength and stability to the disc and withstand compressive forces (*Sivan et al., 2014*; *Pattappa et al., 2012*). The NP plays a crucial role in the progression of IVD degeneration due to its susceptibility to significant structural and functional changes during aging and degeneration. Indeed, NP structural changes are easily detected through imaging techniques. Thus, because of its high water content, MRI has made the NP the most visible indicator of disc degeneration in clinical practice. In contrast, degenerated alterations in the AF are more closely linked to pain and disability. As a result, effective therapeutic strategies for IVD degeneration should aim to restore the functionality of the entire disc unit.

Cellular senescence, triggered by normal cells in response to various intrinsic and extrinsic stressors, is a fundamental mechanism underlying age-related chronic diseases. Senescent cells are featured by irreversible growth arrest and acquire a senescent-associated secretory phenotype (SASP) and the secretion of pro-inflammatory cytokines, chemokines, and tissue-damaging proteases (*Ngo et al., 2017*). Activation of P53/P21 and P16 pathways plays a critical role in regulating senescence (*Reimann et al., 2024*). In addition, the NF-κB pathway is a major pathway involved in the expression of pro-inflammatory mediators in senescent cells (*Salminen et al., 2012*). Factors within the SASP can act in an autocrine manner to reinforce the senescent state and induce senescence in neighboring cells via paracrine signaling (*Kumari and Jat, 2021*). This mechanism elucidates the accumulation of senescent cells with aging and its associated detrimental effects. In the IVD, it has been well established that the number of senescent cells increases with aging and IVD degeneration (*Veroutis et al., 2021*; *Gruber et al., 2007*; *Gruber et al., 2009*), as demonstrated by increased senescence-associated β-galacto-sidase (SA-β-Gal) positive cells and decreased cell proliferation. Growing research also demonstrates that targeting cellular senescence holds promise in alleviating the progression of IVD degeneration. Long-term systemic or local clearance of senescent cells by senolytic drugs mitigated age-related IVD degeneration in mice (*Novais et al., 2021*) and reduced the expression of pro-inflammatory cytokines and matrix proteases and restored IVD structure in an injury-induced IVD degeneration model in rats (*Lim et al., 2022*). Cherif et al. and Mannarino et al. demonstrated that in humans, senolytic drugs o-Vanillin (*Mannarino et al., 2021*), and RG-7112 (*Cherif et al., 2020*) attenuated NP cellular senes-cence, where a combination of these two senolytic drugs reduced the release of inflammatory factors and pain mediators in degenerated NP cells from low back pain patients (*Mannarino et al., 2023*).

PDGF is a major constituent of platelet-rich plasma (PRP), which is widely used in the clinical setting for tissue regeneration and repair. It can be made of a homodimer of A, B, C, and D polypeptide chains, or an AB heterodimer (*Chen et al., 2013*; *Heldin and Lennartsson, 2013*). Among these, PDGF-AB and -BB are the predominant forms in PRPs (*Weibrich et al., 2002*). Two types of PDGFR (PDGFRA and PDGFRB) were identified that differ in ligand-binding specificity. PDGFRA binds to all PDGF chains but D chain while PDGFRB only binds to B and D chains (*Chen et al., 2013*; *Heldin and Lennartsson, 2013*). Upon binding, the PDGF-PDGFR network induces a cascade of phosphor-ylation in pathways including cell proliferation, migration, and survival (*Contreras et al., 2021*). We previously demonstrated that recombinant human PDGF-BB (rhPDGF-BB) inhibited cell apoptosis while promoting cell proliferation and matrix production in human IVD cells (*Presciutti et al., 2014*). Furthermore, when delivered via hydrogel, rhPDGF-BB successfully inhibited IVD degeneration and restored IVD biomechanical function in a rabbit preclinical model of puncture-induced IVD degen-eration (*Paglia et al., 2016*). In the current study, we hypothesize that PDGF-AB and -BB mitigated IVD degeneration by targeting cellular senescence. To test it, aged, degenerated human IVD cells were treated with rhPDGF-AB or -BB and transcriptome profiling was performed on treated NP and AF cells. We found similar but distinct responses to the treatment between NP and AF cells. In NP

**Table 1.** Donor information.

| Grade | Age | Sex | Cell type |
| --- | --- | --- | --- |
| 4 | 67 | M | NP & AF |
| 5 | 61 | M | NP & AF |
| 5 | 61 | M | NP & AF |
| 4 or 5 | 65 | F | NP & AF |
| 4 or 5 | 81 | F | AF |
| 4 | 53 | F | AF |
| 4 | 64 | F | NP |
| 1 | 19 | M | NP & AF |
| 1 | 21 | M | NP & AF |
| 1 | 25 | F | NP & AF |
| 1 | 27 | F | NP & AF |

cells, PDGF-AB and -BB treatment upregulated the expression of PDGFRA and genes involved in cell cycle progression while downregulating the expression of genes related to ROS production, oxidative stress, and mitochondrial function. The alterations in these pathways indicated that the senescent phenotype was alleviated in human degenerated IVD cells treated with PDGF-AB/BB. Furthermore, in human senescent NP cells induced by irradiation, PDGFRA was significantly reduced, while PDGF-AB/ BB treatment mitigated the progression of senescence through increased cell cycle progression and reduced SA-β-Gal staining and the expression of P21, P16, NF-κB, and IL6.

## Methods

### Sample collection and cell extraction

This study was conducted in accordance with the ethical principles outlined in the Declaration of Helsinki. All experiments and procedures were reviewed and approved by the institutional review board of Emory University (IRB #00099028) approval. Human degenerated NP and AF tissues (Grade IV or V on Pfirrman grade; 64.6±8.5 y old) were obtained as the surgical waste from donors with disc herniation, with each donor providing written informed consent. Healthy NP and AF cells (23.0±3.7 y old) were gifted by Professor Lisbet Haglund from McGill University (Tissue Biobank #2019–4896). The donor information was listed in *Table 1*. NP and AF cells were extracted as previously described (*Zhang et al., 2024*). Isolated cells were expanded in growth media containing low glucose DMEM, 10% FBS, 1 X antibiotic-antimycotic (Thermo Fisher Scientific, 15240062), and 50 µg/ml L-Ascorbic acid 2-phosphate (Sigma, A8960). Cells were grown at 37 °C under 5% $CO_2$ and 20% $O_2$. To minimize the variability between experiments that may be caused by changes in cell behavior with each passage, cells of passage 3 were used for each experiment.

### PDGF treatment in human intervertebral disc cells

To synchronize cells and enhance the sensitivity of cells to PDGF stimulation, degenerated NP (n=5/ each group) and AF cells (n=6/each group) were serum-deprived in low glucose media containing 0.2% FBS for 1 d. Cells were then treated with recombinant human PDGF-AB (40 ng/ml; PeproTech, 10770584) or -BB (20 ng/ml; PeproTech, 10771918) for 5 d.

### RNA-sequencing

Total RNA was isolated with TRIzol (Thermo Fisher Scientific, 15596018) and purified using miRNeasy kit (Qiagen, 217084). Genomic DNA contamination was eliminated through on-column DNase digestion. The concentration and quality of RNA were determined using Nanodrop and RNA integrity was assessed by Agilent 2200 Bioanalyzer. RNA samples (NP: n=5, AF: n=6) with RIN >7 were shipped in dry ice to Novogene for RNA sequencing, which was carried out using NovaSeq 6000 (Novogene Corporation Inc). Each sample represents a unique biological replicate, and no technical replicates were performed for RNA sequencing. Libraries were constructed from 0.2 µg RNA using the TruSeq Stranded Total RNA Sample Prep Kit (Illumina). Quality control was performed for the distribution of sequencing error rate and GC content and data filtering was performed to remove the low-quality reads and reads containing adapters. The resulting sequences were uploaded into the NCBI Sequence Read Archive (SRA) database (PRJNA1150962). The clean reads were mapped to the human reference genome sequence using DNASTAR.

### Differential gene expression analysis of RNA-seq data

Unnormalized raw counts exported from DNASTAR were used for pairwise gene expression comparison between untreated and PDGF-AB/BB samples. Differential expression genes (DEGs) were identified using the DESeq2 package in R and DEGs with a fold change greater than 1.5 and adjusted p value less than 0.05 were considered statistically significant. Heatmap and volcano plots and principal component analysis were performed to visualize the DEGs between untreated and PDGF-AB/ BB samples.

### Pathway enrichment analysis

To identify the biological process that DEGs were enriched, Gene ontology (GO) analysis was performed using upregulated and downregulated DEGs separately in R. The significantly enriched

GO terms were visualized by dot plot analysis. GeneRatio in the X-axis was calculated by the ratio of gene counts enriched in a particular GO term to all the gene counts annotated in the GO term.

Gene Set Enrichment Analysis (GSEA) was performed on all the genes to determine whether a predefined set of genes is significantly enriched in untreated or PDGF-BB samples. GSEA was conducted based on the KEGG (Kyoto Encyclopedia of Genes and Genomes) database using the clusterProfiler package in R (*Wu et al., 2021*) and a gene set was considered significantly enriched when the false discovery rate (FDR) was less than 0.05. Enrichment score (ES) was calculated to reflect the degree to which a gene set is overrepresented in our dataset, which was ranked based on fold change. Normalized enrichment score (NES) is the ES normalized to the gene set size.

## Protein-protein interaction network analysis

The interactions among DEGs were performed using the Search Tool for the Retrieval of Interacting Genes (STRING) database (v12.0) with an interaction score greater than 0.4. Discrete proteins were removed, and the node files were exported and visualized in Cytoscape (v3.10.0). To identify essential nodes in the interaction network, the betweenness centrality among the nodes was calculated using the CytoNCA plugin (v2.1.6) in Cytospace (*Tang et al., 2015*) and the top 10 hub nodes were identified using the CyoHubba plugin (v0.1) (*Chin et al., 2014*).

## Cellular senescence induction by irradiation

To induce cellular senescence, cells were seeded in a six-well plate at a density of $5 \times 10^4$ cells/ml and cultured in low glucose DMEM supplemented with 10% FBS, 1 X antibiotic-antimycotic (Thermofisher Scientific, 15240062), and 50 µg/ml L-Ascorbic acid 2-phosphate (Sigma, A8960). After 48 hr of culture, growth media was removed, and cells were washed with 1 X PBS once. To minimize the interference of culture media with the irradiation process, 500 µl PBS was added before irradiation. Healthy NP and AF cells (n=4/each group) were exposed to a single dose of 5 Gy, 10 Gy, or 15 Gy of X-ray in a CellRad benchtop X-ray irradiator (Precision X-Ray) at a rate of 2.6 Gy/min. The irradiation doses and rate were selected based on prior studies (*Ungvari et al., 2013*; *Zhong et al., 2024*) and optimized to effectively induce senescence-associated changes in human IVD cells without triggering excessive cell death, which is often observed at higher irradiation doses. After irradiation, cells were immediately incubated with growth media. Cells exposed to 10 Gy of irradiation showed the most pronounced senescent phenotype on day 10 and were used for further experiments. For treatment experiments, cells were exposed to 10 Gy of X-ray and treated with PDGF-AB/BB (20 ng/ml) immediately after irradiation for 10 d. The culture of media was changed every other day.

## Senescence-associated β-galactosidase (SA-β-Gal) staining

SA-β-Gal staining was performed according to the manufacturer's protocol (Cell Signaling Technology, #9860). In brief, cells cultured in six-well plates were washed in PBS and fixed with 1 X fixative solution for 10 min. 1 ml of β-Gal staining solution (containing 1 mg/ml X-Gal) was added and the cells were incubated at 37 °C in the absence of $CO_2$ overnight. Cells were washed in PBS and observed under a bright-field microscope (ECHO Revolve). Technical triplicates were performed for each sample.

## Immunocytochemistry

Cells cultured on glass coverslips were washed in 1 X PBS and fixed in 4% (v/v) formaldehyde for 10 min. After washing in 1 X PBS, cells were premetallized in PBS with 0.1% Triton X-100 for 15 min. Non-specific binding sites were blocked in 0.1% Tween-20/PBS with 1% bovine serum albumin for 30 min, followed by incubating with primary antibodies against P21 (1:500, Cell Signaling Technology, #2947), P16 (1:500, Cell Signaling Technology, #80772), or Lamin B1 (1:500, Cell Signaling Technology, #68591 S). After overnight incubation at 4 °C, the cells were washed and incubated with the corresponding host-specific secondary antibodies goat anti-mouse Alex Fluor 488 (1:1000, #ab150113), or goat anti-rabbit Alex Fluro 647 (1:1000, #ab150079) for 1 hr. Cells were counterstained and mounted with mounting medium with DAPI (Abcam, #ab104139) and visualized using an Olympus BX63 automated fluorescence microscope. Technical triplicates were performed for each sample.

**Table 2.** Sequence of primers.

| Gene | Primer sequence |
| --- | --- |
| P21 | Forward: 5'-GAC ACCACT GGA GGG TGA C T-3'<br>Reverse: 5'-CAGGTC CAC ATG GTC TTC CT-3' |
| P16 | Forward: 5'-CCAACGCACCGAATAGTTACG-3'<br>Reverse: 5'-GCGCTGCCCATCATCATG-3' |
| IL6 | Forward: 5'-CCGGGAACGAAAGAGAAGCT-3'<br>Reverse: 5'-GCGCTTGTGGAGAAGGAGTT-3' |
| CCNB1 | Forward: 5'-ACTGGGTCGGGAAGTCACTG-3'<br>Reverse: 5'-CATTCTTAGCCAGGTGCTGC-3' |
| CCND1 | Forward: 5'-CTGTGCTGCGAAGTGGAAAC-3'<br>Reverse: 5'-TCTGTTTGTTCTCCTCCGCC-3' |
| CASP3 | Forward: 5'-TGG TTC ATC CAG TCG CTT TG-3'<br>Reverse: 5'-ATT CTG TTG CCA CCT TTC G-3' |
| MKI67 | Forward: 5'-ATTTGCTTCTGGCCTTCCCC-3'<br>Reverse: 5'-CCAAACAAGCAGGTGCTGAG-3' |
| ACTB | Forward: 5'-CTC TTC CAG CCT TCC TTC CT-3'<br>Reverse: 5'-AGC ACT GTG TTG GCG TAC AG-3' |

## Quantitative real-time PCR assays

Total RNA was extracted from cells collected in TRIzol reagent (Thermo Fisher Scientific, #15596018) and RNA quality was measured using a NanoDrop-1000 spectrophotometer (Thermo Fisher Scientific, #ND-2000). 1 µg of RNA was reverse transcribed using Verso cDNA synthesis kit (Thermo Fisher Scientific, #AB-1453B). Quantification of mRNA expression was determined by SYBR-based real-time PCR using PowerUp SYBR Green Master Mix (Applied Biosystems, #A25742). PCR primer sequences were listed in *Table 2*. Values were normalized to *ACTB*. mRNA levels were presented as fold change compared to untreated cells according to the $2^{-\Delta\Delta CT}$ method. Each qPCR assay was performed in technical triplicates for each sample.

## Immunoblotting

Whole-cell protein extraction and immunoblotting analysis were performed as previously described (*Zhang et al., 2024*). Briefly, cells were lysed using RIPA lysis buffer with 1 X protease and phosphatase inhibitor (Thermo Fisher Scientific, #PI78440). Protein concentration was determined using a Pierce BCA protein assay kit (Thermo Fisher Scientific, #23225) and 20 µg of cell lysate from each sample was used to determine the protein expression of P21 (1:2000, Cell Signaling Technology, #2947), NF-kB (1:2000, Cell Signaling Technology, #8242 S), and PDGFRα (1:1000, Cell Signaling Technology, #3164 S). All proteins were normalized to β-actin (1:2000, Cell Signaling Technology, #5125 S). Technical triplicates were performed for each sample.

## Statistics

One-way analysis of variance (ANOVA) testing was performed to assess the effects of treatment on NP and AF cells with subsequent Dunnett post hoc testing. To analyze the effects of ionizing radiation on the expression of senescence markers in cells, a t-test was utilized. Data are presented as mean ± standard deviations. There were no samples were excluded from the analysis. The statistical analyses were performed using GraphPad Prism 9. A p-value < 0.05 was considered statistically significant.

## Results

### Transcriptomics of AF and NP cells in response to PDGF-AB/BB

Bulk RNA sequencing was performed to investigate the global changes in gene expression of degenerated AF and NP cells in response to rhPDGF-AB/BB treatment. Principal component analysis (PCA) on all detected genes revealed distinct clusters between untreated and rhPDGF-AB/BB-treated

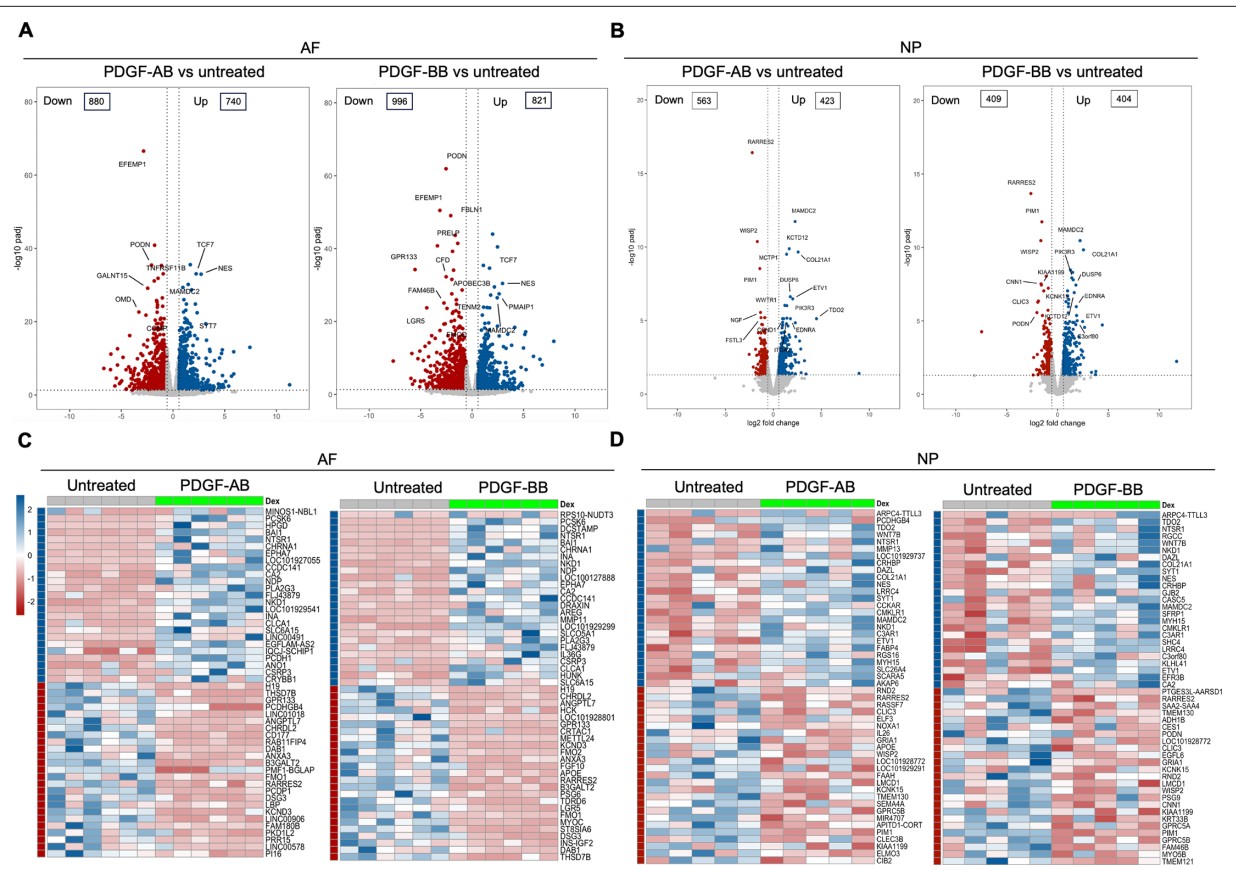

**Figure 1.** Transcriptomics of annulus fibrosus (AF) and nucleus pulposus (NP) cells in response to platelet-derived growth factor (PDGF)-AB/BB treatment. (**A, B**) Volcano plots showing the differentially expressed genes (DEGs) in AF (**A**) and NP (**B**) cells treated with rhPDGF-AB/BB for 5 d (cutoff: |FC|>1.5 and FDR < 0.05). DEGs were annotated in the plot according to false discovery rate (FDR). (**C, D**) Heatmap plots showing the top 50 DEGs in rhPDGF-AB/BB-treated AF (**C**) and NP (**D**) cells based on fold change. NP: n=5 samples. AF: n=6 samples.

The online version of this article includes the following figure supplement(s) for figure 1:

**Figure supplement 1.** Principle component analysis (PCA) of nucleus pulposus (NP) and annulus fibrosus (AF) samples treated with rhPDGF-AB and BB.

samples, while clusters of rhPDGF-AB and rhPDGF-BB-treated samples were overlapping (*Figure 1—figure supplement 1*). As shown in volcano plots, differential expression analysis (|FC|>1.5 and FDR <0.05) revealed 880 downregulated and 740 upregulated genes in AF cells treated with rhPDGF-AB and 996 downregulated and 821 upregulated genes induced by rhPDGF-BB treatment after 5 d of exposure (*Figure 1A*). In contrast, NP cells had 563 downregulated and 423 upregulated genes in rhPDGF-AB-treated group and 409 downregulated and 404 upregulated genes in rhPDGF-BB-treated group. The top differentially expressed genes (DEGs) were annotated according to FDR and most of these genes exhibited similarity between treatment groups in both AF and NP cells (*Figure 1A and B*). Furthermore, a heat map visualizing the top 50 DEGs by fold change revealed variability among donors while demonstrating an overall consistent response to rhPDGF-AB/BB treatment across the samples (*Figure 1C and D*). Tissue-specific changes were observed in AF and NP cells when treated with rhPDGF-AB/BB. For example, *SLC26A4* (Pendrin) and *CA2*, regulators of the process of transporting and producing bicarbonate, are involved in the homeostatic control of pH in various types of cells (*Vince and Reithmeier, 1998*; *Silagi et al., 2018*) and their expression was upregulated in NP cells when treated with rhPDGF-AB/BB (*Figure 1D*). *MAMDC2* and *COL21A1*, involved in extracellular matrix organization (*Chou and Li, 2002*; *Mavillard et al., 2023*), were also upregulated in NP cells, while *WISP2* (*Ruiz-Fernández et al., 2022*) and *RARRES2* (*Liu-Chittenden et al., 2017*), modulators of the induction of inflammatory mediators and immune response, were downregulated after treatment. In contrast, in addition to the commonly upregulated *CA2, MDMC2* and *RARRES2*, AF cells showed alterations in genes involved in mechanical stress, pain sensation, and angiogenesis, such as *CSRP3*

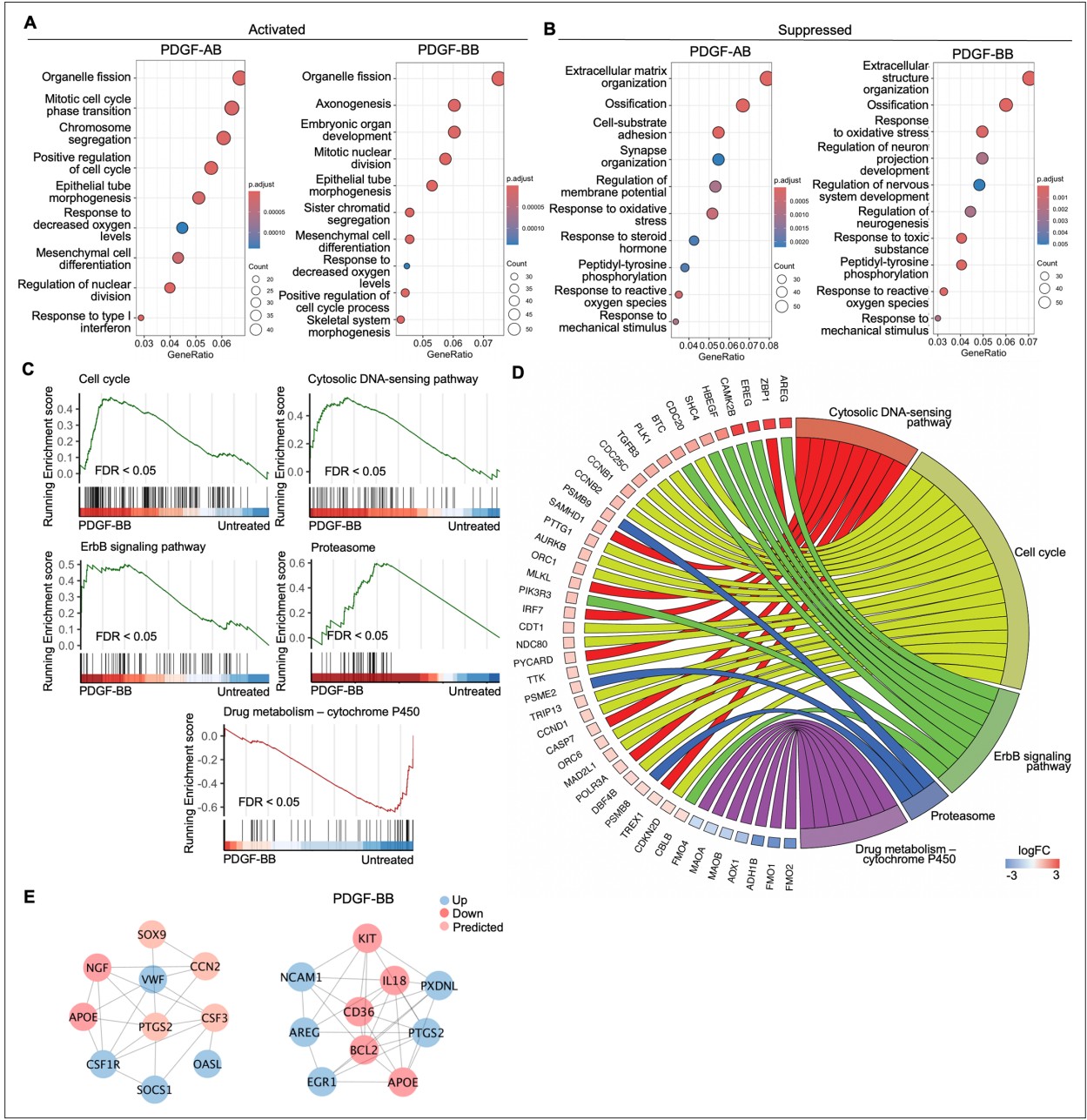

**Figure 2.** Functional analysis of annulus fibrosus (AF) cells treated with platelet-derived growth factor (PDGF)-AB/BB. (**A, B**) Gene ontology (GO) analysis for biological process using upregulated (**A**) and downregulated (**B**) differential expression genes (DEGs) in AF cells. Y-axis label represents gene ontology (GO) terms, and x-axis represents geneRatio (geneRatio refers to the proportion of genes that are annotated in specific GO term within all the input genes). The size of the bubble represents the gene counts enriched in a particular GO term and the color indicates the false discovery rate (FDR) of GO terms. (**C**) Gene set enrichment analysis (GSEA) on the entire transcriptome profile demonstrated an upregulation of the cell cycle, cytosolic DNA-sensing pathway, ErbB signaling pathway, and proteasome and a downregulation of drug metabolism-cytochrome P450 pathway in rhPDGF-BB-treated NP cells. (**D**) The leading edge genes associated with these pathways were shown using a chord diagram. (**E**) Protein-protein interaction network using DEGs was constructed in STRING and visualized in Cytoscape.

(*Rashid et al., 2015*), PI16 (*Singhmar et al., 2020*), *EFEMP1* (*Song et al., 2011*), and *SLC26A4-AS1* (*Li et al., 2021*; *Figure 1C*). Notably, *PRELP*, potentially associated with Hutchinson-Gilford progeria (*Lewis, 2003*), was also downregulated in rhPDGF-AB/BB-treated AF cells (*Figure 1A*).

## Functional analysis of AF cells treated with PDGF-AB/BB

To investigate the enriched biological processes (BP) induced by rhPDGF-AB/BB treatment in AF cells, gene ontology (GO) analysis was performed using upregulated and downregulated DEGs separately. As expected, the biological processes between rhPDGF-AB and -BB-treated samples were mostly overlapping (*Figure 2A and B*). In AF cells, upregulated DEGs were significantly enriched in biological processes including regulation of cell cycle and chromosome segregation. Furthermore, AF cells demonstrated upregulated pathways including mesenchymal cell differentiation and response to oxygen levels and downregulated pathways including response to ROS and oxidative stress (*Figure 2A*). Moreover, the treatment downregulated groups of genes related to neurogenesis and response to mechanical stimulus (*Figure 2B*). In line with the results of GO analysis, GSEA analysis revealed an overrepresentation of cell cycle pathway in rhPDGF-AB/BB-treated cells (*Figure 2C*). Additionally, the ErbB signaling pathway, proteasome pathway, and cytosolic DNA-sensing pathway were enriched, suggesting that PDGF-AB/BB treatment promoted cell growth, protein turnover, and maintenance of cellular homeostasis. The leading-edge genes that contributed most significantly to these pathways include *CCNB1, CCND1, AREG, PSMB9, TREX1*, and *POLR3A*, as shown in *Figure 2D*. Protein-protein interaction (PPI) network was employed to further investigate the potential connectivity of DEGs between untreated and treated groups using the STRING and Cytoscape softwares (*Figure 2E*). PPI analysis revealed that the hub genes in rhPDGF-AB-treated samples were associated with inflammation (*CCN2, CSF3, OASL, and SOCS1*), macrophage function (*CSF1R*), neurogenesis (NGF, APOE), and angiogenesis (PTGS2). In rhPDGF-BB-treated AF cells, *PXDNL,* a member of the peroxidase enzyme family and participating in scavenging ROS interacted with *IL-18*, an activator of NF-kB signaling. Additionally, *CD36*, involved in angiogenesis, was among the top 10 hub genes. Taken together, these findings suggest that PDGF treatment promoted cell mitogenesis while inhibiting oxidative stress, inflammation, neurogenesis, and response to mechanical stimulus.

## Functional analysis of NP cells treated with PDGF-AB/BB

GO analysis revealed similar yet distinct responses to rhPDGF-AB/BB treatment between NP and AF cells. In NP cells**,** rhPDGF-AB and -BB treatment upregulated groups of genes related to cartilage development, cell-matrix adhesion, Wnt signaling pathway, and response to decreased oxygen levels, which were all important to maintaining NP phenotype (*Figure 3A*). Downregulated DEGs were enriched in GO terms associated with fatty acid metabolism, cellular respiration, mitochondrial function, and reactive oxygen species production (*Figure 3B*), suggesting alterations in cellular metabolism and oxidative stress. Gene set enrichment analysis (GSEA) on the entire transcriptomics also revealed that Wnt signaling pathway and neuroactive ligand-receptor interaction were upregulated, while the pathways related to oxidative phosphorylation and ribosome were downregulated in rhPDGF-BB-treated NP cells (*Figure 3C*). The core enrichment genes associated with these pathways were visualized in a chord diagram (*Figure 3D*). Specifically, the leading-edge DEGs enriched in Wnt signaling pathway include *WNT7B, WNT5A, NKD1*, and *SFRP1* and oxidative phosphorylation pathway showed enrichment in *NDUFB7, NDUFA11, COX5B*, and *CYC1*.

To better understand the regulatory mechanisms of PDGF-AB/BB in degenerated NP cells, DEGs between untreated and treated groups were used to construct PPI networks. The network of the top 10 hub nodes based on betweenness was demonstrated in *Figure 3E* and it revealed the strong correlation between *PDGFRA (*PDGF receptor alpha) and *IL6, MDM2, MDM2, CCND1, PIK3R1, NOP53, FN1,* and *IDH2*. The enrichment of hub gene *IDH2*, which plays a role in generating NADPH within mitochondria, further validated the alterations in oxidative phosphorylation pathway and subsequent ROS production, as indicated by GSEA and GO analysis in rhPDGF-AB/BB-treated NP cells. *MDM2* (a negative regulator of P53) and predicted *NOP53* (a regulator of P53 binding activity) suggested that PDGF-BB interfered with P53 signaling pathway. The interaction between these genes and inflammatory cytokine *IL6* indicated a potential involvement of PDGF-AB/BB in cellular senescence. The gene expression of top hub genes PDGFRA and IL6 was further verified in both NP (*Figure 3F*) and AF (*Figure 3G*) cells by qPCR. PDGF-AB/BB increased the gene expression of PDGFRA in only NP cells

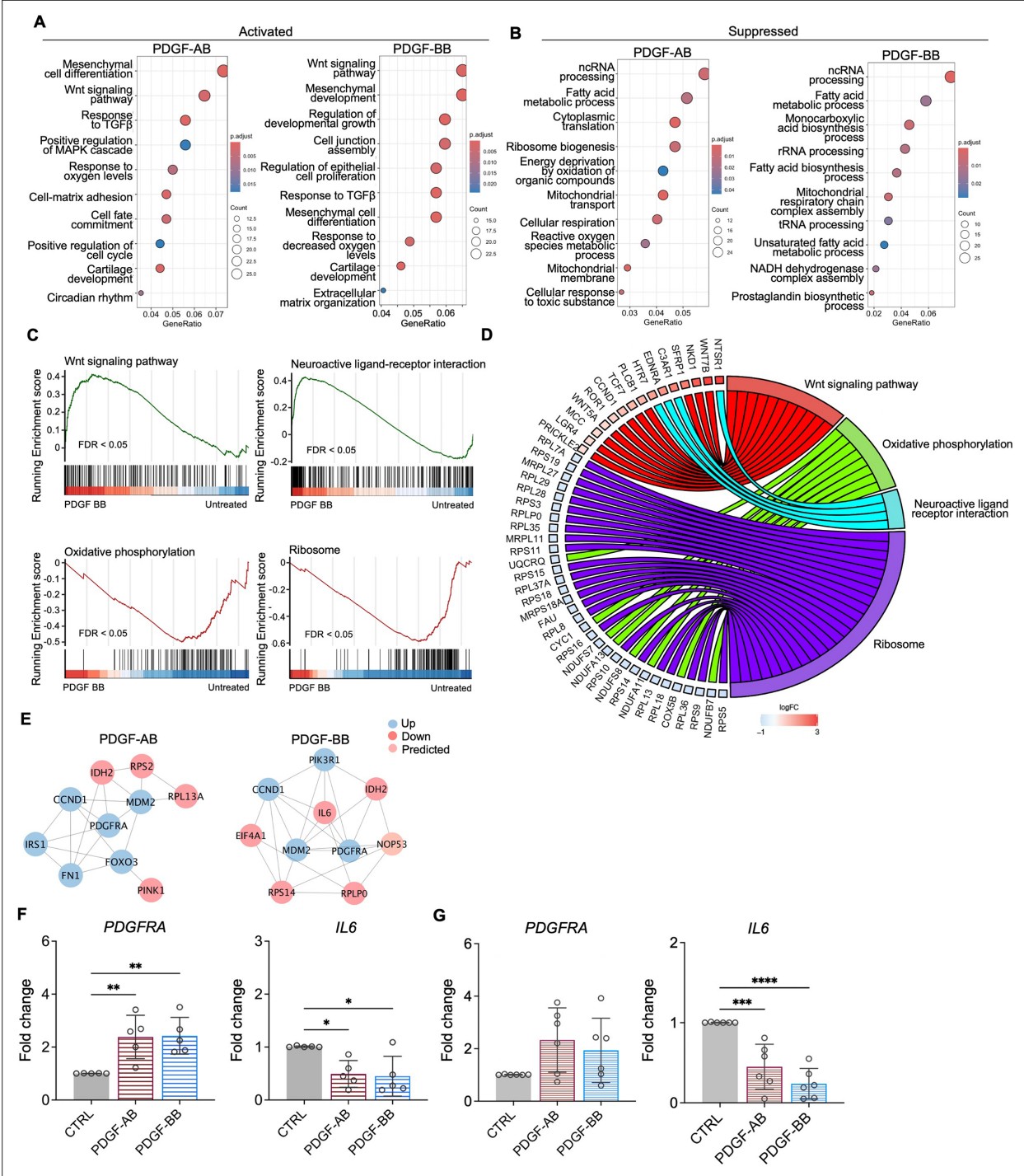

**Figure 3.** Functional analysis of nucleus pulposus (NP) cells treated with platelet-derived growth factor (PDGF)-AB/BB. (**A, B**) Gene ontology (GO) analysis for biological process using upregulated (**A**) and downregulated (**B**) differential expression genes (DEGs) in NP cells. Y-axis label represents GO terms, and x-axis represents geneRatio (geneRatio refers to the proportion of genes that are annotated in specific GO term within all the input genes). The size of the bubble represents the gene counts enriched in a particular GO term and the color indicates the false discovery rate (FDR) of GO terms. (**C**) Gene set enrichment analysis (GSEA) on the entire transcriptome profile demonstrated an upregulation of Wnt signaling pathway and a downregulation of oxidative phosphorylation and ribosome in rhPDGF-BB-treated NP cells. (**D**) The leading-edge genes associated with these pathways were shown using a chord diagram. (**E**) Protein-protein interaction network using DEGs was constructed in STRING and visualized in Cytoscape. (**F, G**) The changes in the gene expression of PDGFRA and IL6 were verified in NP (**F**) and annulus fibrosus (AF) (**G**) cells treated with rhPDGF-AB (40 ng/ml) or -BB (20 ng/ml) for 5 d. One-way ANOVA testing was performed to assess the effects of treatment on NP and AF cells with subsequent Dunnett post hoc testing. Data are presented as mean with SD. NP: n=5; AF: n=6. *p<0.05. **p<0.01. ***p<0.001. ****p<0.0001.

*Figure 3 continued on next page*

*Figure 3 continued*

The online version of this article includes the following source data and figure supplement(s) for figure 3:

**Source data 1.** Excel file containing the measures displayed in *Figure 3*.

**Figure supplement 1.** Protein expression of PDGFRA was decreased in nucleus pulposus (NP) and annulus fibrosus (AF) samples were treated with rhPDGF-AB and BB.

**Figure supplement 1—source data 1.** Excel file containing the measures of PDGFRA protein expression levels in untreated and platelet-derived growth factor (PDGF)-treated disc cells.

**Figure supplement 1—source data 2.** PDF file containing original western blots for *Figure 3—figure supplement 1*, indicating the bands and treatments.

**Figure supplement 1—source data 3.** Original files for western blots displayed in *Figure 3—figure supplement 1*.

and reduced the IL6 gene expression in both types of cells. Interestingly, the protein expression of PDGFRA was decreased in treated NP and AF cells compared to untreated groups (*Figure 3—figure supplement 1*). The discrepancy in changes between gene and protein expression levels reflects the complexity in the dynamic regulation of PDGFRA (*Figure 3—figure supplement 1*). Taken together, NP and AF cells responded differently to PDGF-AB/BB treatment, but both cell types showed a positive regulation of cell cycle and inhibited inflammation, response to ROS, oxidative stress, and mitochondrial dysfunction, indicating a role of PDGF-AB/BB in IVD cell senescence.

## Induction of NP cellular senescence through X-ray radiation

To test our hypothesis that PDGF-AB/BB provides protection against the progression of cellular senescence in the IVD, we first created a cellular senescence model using X-ray radiation. The healthy NP cells were exposed to 5, 10, and 15 Gy of irradiation and cultured for 7–10 d to develop a senescent phenotype. SA-β-gal staining was performed to detect senescent cells. We found that non-irradiated, proliferating NP cells showed elongated, spindle-shaped morphology and faint SA-β-gal staining at day 7 (*Figure 4A*). In contrast, cells exposed to 5 and 10 Gy of irradiation developed characteristics of a senescent cell phenotype — enlarged cell size, low cell number, and a higher percentage of SA-β-gal positive cells. However, cells exposed to 15 Gy of radiation underwent significant cell death, showing negligible SA-β-gal staining. As a result, we excluded the 15 Gy dosage group in subsequent studies. To quantify cell growth, the MTT assay was performed, which revealed reduced cell proliferation under 5 and 10 Gy of irradiation compared to the non-irradiated group (*Figure 4B*). Furthermore, we examined the expression of senescence-related markers — Lamin B1, P21, and P16. Lamin B1 is a structural component of the nucleus, and its loss is a senescence-associated biomarker (*Freund et al., 2012*). Non-irradiated cells displayed nuclear Lamin B1 expression, while its expression was lost in healthy NP cells exposed to 5 and 10 Gy of irradiation for 7 d (*Figure 4C*). Meanwhile, nuclear localization of P21 (*Figure 4C*) and P16 (*Figure 4D*) was observed in irradiated cells, with little immunopositivity in non-irradiated cells at day 7. However, the transcripts of *P21* and *P16* were not significantly decreased until day 10 (*Figure 4E and F*). Caspase 3 (*CASP3*) gene expression was reduced at both timepoints, suggesting that the irradiated cells were resistant to apoptosis. Furthermore, the gene expression of *PDGFRA* was significantly decreased in cells under 10 Gy of irradiation at day 10. Taken together, when exposed to irradiation for 10 d, healthy human NP cells developed a senescent phenotype, accompanied by decreased PDGFRA expression.

## PDGF-AB/BB inhibited senescent phenotype in NP and AF cells

To investigate the protective effects of PDGF-AB/BB against cell senescence in the IVD, healthy human NP and AF cells were exposed to 10 Gy of irradiation, followed by immediate treatment with 20 ng/ml rhPDGF-AB/BB for 10 d. We first examined the cell cycle progression using DAPI-stained cells and found that rhPDGF-BB significantly increased the percentage of cell population in the S phase in irradiated NP (*Figure 5A and B*) and AF (*Figure 5C and D*) cells at day 10. In rhPDGF-AB-treated cells, the percentage of cell population in the S phase was increased in AF cells, but not significantly in NP cells. These findings suggested that PDGF-AB/BB rescued IVD cells from cell cycle arrest in irradiation-induced senescent cells. Additionally, SA-β-gal staining demonstrated an increase in the number of senescent cells after irradiation at day 10, whereas a decrease was examined in rhPDGF-AB/BB-treated NP and AF cells (*Figure 5E*).

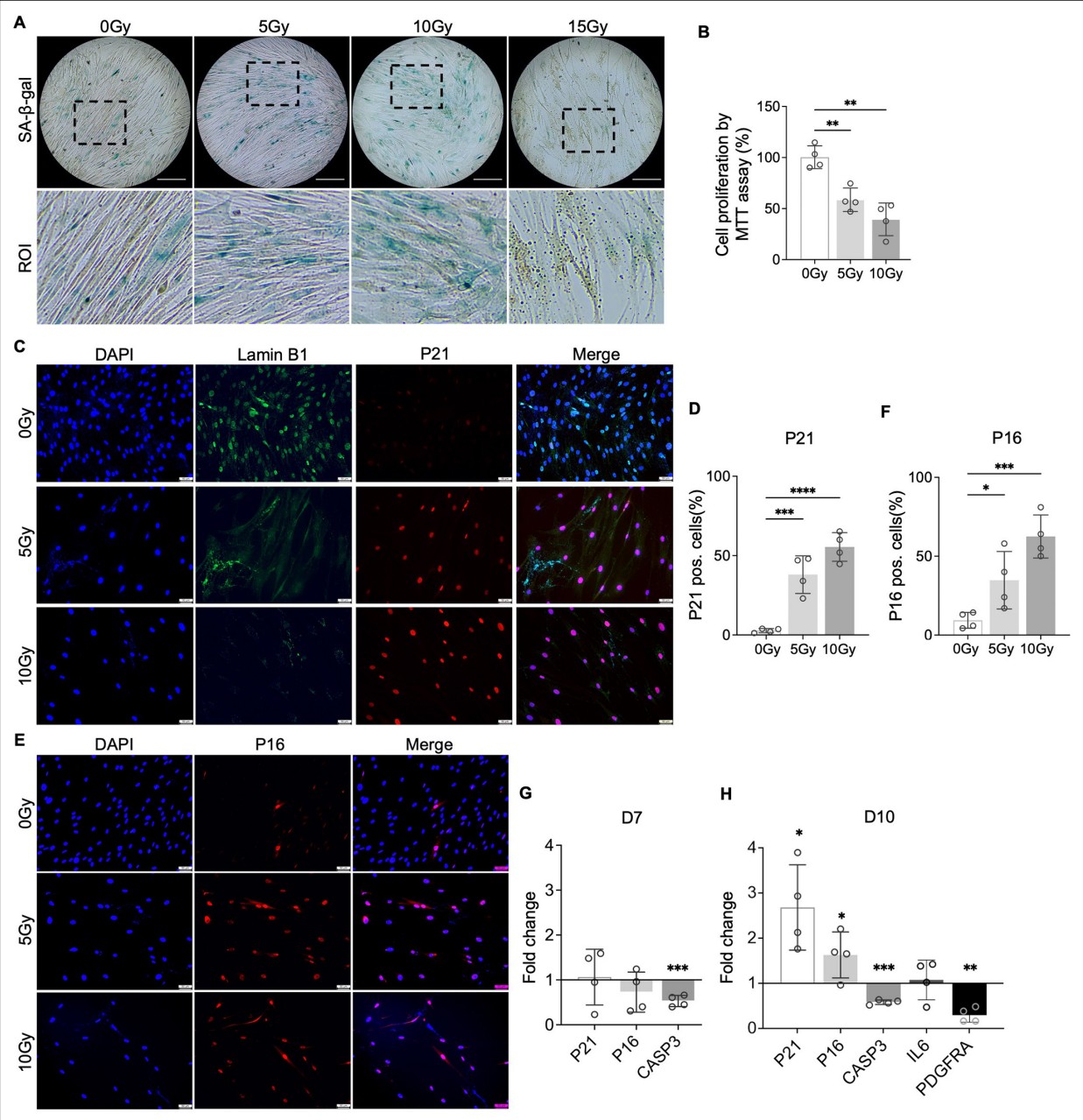

**Figure 4.** Cellular senescence induction by irradiation. (**A**) Senescence-associated β-galactosidase (SA-β-gal) staining in healthy nucleus pulposus (NP) cells under different doses of irradiation at day 7. Images were captured under 10 X magnification using Echo microscope. ROI: region of interest. Scale bar = 200 μm. (**B**) MTT assay showed decreased cell proliferation after irradiation at day 7. One-way ANOVA testing was performed to assess the effects of irradiation dosage on NP and annulus fibrosus (AF) cells with subsequent Dunnett post hoc testing. Data are presented as mean with SD. n=4. **p<0.01. (**C**) Immunocytochemistry staining on Lamin B1 and P21 in irradiated cells at day 7. Scale bar = 50 μm. (**D**) Quantification of P21-positive cells. One-way ANOVA testing was performed to assess the effects of irradiation dosage on NP and AF cells with subsequent Dunnett post hoc testing. Data are presented as mean with SD. n=4. ***p<0.001. ****p<0.0001. (**E**) Immunocytochemistry staining on P16 in irradiated cells at day 7. Scale bar = 50 μm. (**F**) Quantification of P16 positive cells. One-way ANOVA testing was performed to assess the effects of irradiation dosage on NP and AF cells with subsequent Dunnett post hoc testing. Data are presented as mean with SD. n=4. *p<0.05. ***p<0.001. (**G**) Changes in the gene expression levels of *P21*, *P16*, and *CASP3* in NP cells under 10 Gy of irradiation compared to the non-irradiated group at day 7 and 10 timepoints. (**H**) The gene expression of *PDGFRA* was reduced under 10 Gy of irradiation compared to the non-irradiated group at day 10. t-test was utilized to analyze the effects of ionizing radiation on the expression of senescence markers in cells (**G, H**). Data are presented as mean with SD. n=4. *p<0.05. **p<0.01. ***p<0.001.

The online version of this article includes the following source data for figure 4:

**Source data 1.** Excel file containing the measures displayed in *Figure 4*.

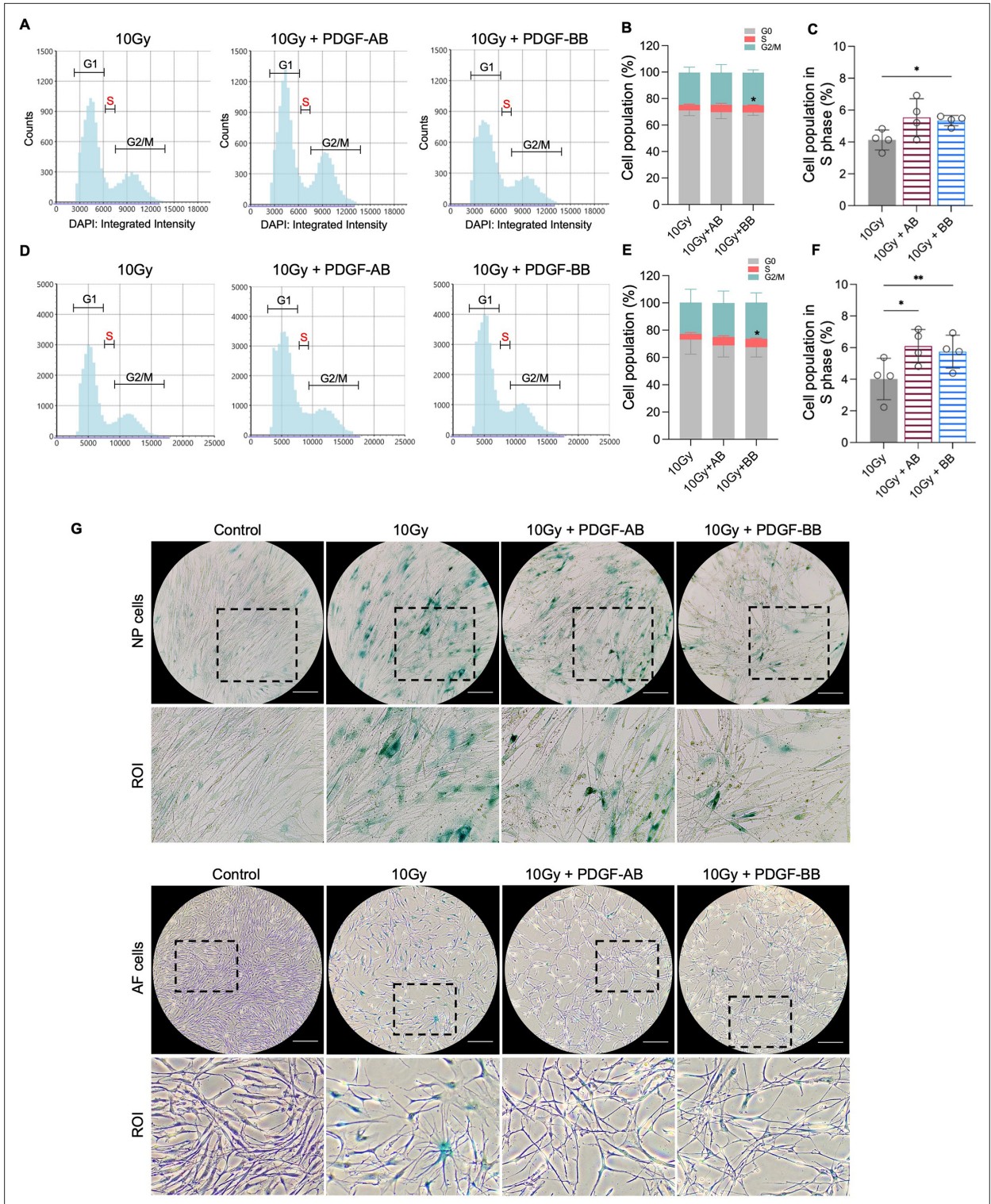

**Figure 5.** Platelet-derived growth factor (PDGF)-AB/BB treatment inhibited the progression of senescence in healthy nucleus pulposus (NP) and annulus fibrosus (AF) cells exposed to irradiation. (A-F) Cell cycle analysis by DAPI staining in irradiated healthy NP (A–C) and AF (D–F) cells treated with rhPDGF-AB (40 ng/ml) or BB (20 ng/ml) for 10 d. The upper panel shows the data obtained from NP cells, and lower panel shows the data from AF cells. Representative graphs (A, D) showing the changes in cell cycle progression in NP and AF cells after treatment. (B, E) Quantification of cell cycle analysis showed the cell percentage in each phase. Two-way ANOVA with Dunnett post hoc testing was performed. n=4 each group. Data are presented as mean with SD. *p<0.05. (C, F) The cell percentage in the S phase was shown to examine the effects of treatment on cell proliferation. One-way ANOVA with Dunnett post hoc testing was performed. n=4 each group. Data are presented as mean with SD. *p<0.05. **p<0.01. (G) SA-β-gal staining in

*Figure 5 continued on next page*

*Figure 5 continued*

irradiated healthy NP (upper panel) and AF cells (lower panel) showing the reduced senescence-associated β-galactosidase (SA-β-gal) positive cells after treatment. Images were captured under 10 X magnification using the Echo microscope. ROI: region of interest. Scale bar = 200 μm.

The online version of this article includes the following source data for figure 5:

**Source data 1.** Excel file containing the measures displayed in *Figure 5*.

At day 10, we analyzed the expression of *PDGFRA* and key senescence regulators of cell cycle inhibitors *P21* and *P16*, *CASP3*, and SASP mediator *IL6*. The treatment of rhPDGF-BB to irradiated NP cells significantly increased the gene expression of *PDGFRA* and decreased the gene expression of *P21*, *P16*, and *IL6*, while playing no effect in the gene expression of *CASP3* (*Figure 6A*). Comparably, the treatment of rhPDGF-AB to irradiated NP cells significantly decreased the gene expression of *P21* and *IL6* and played no effect on the gene expression of *PDGFRA*, *P16*, and *CASP3* (*Figure 6A*). In AF cells, the results show that the treatment of rhPDGF-BB and rhPDGF-AB reduced the gene expression of *P21* to those of NP cells (*Figure 6B*). However, the treatment of either rhPDGF-BB or rhPDGF-AB had no effect on the gene expression of other genes examined (*Figure 6B*). The lack of change in the gene expression of *PDGFRA* from the treatment of either rhPDGF-BB or rhPDGF-AB suggested that PDGF-BB reduced senescent phenotype through other receptors in AF cells.

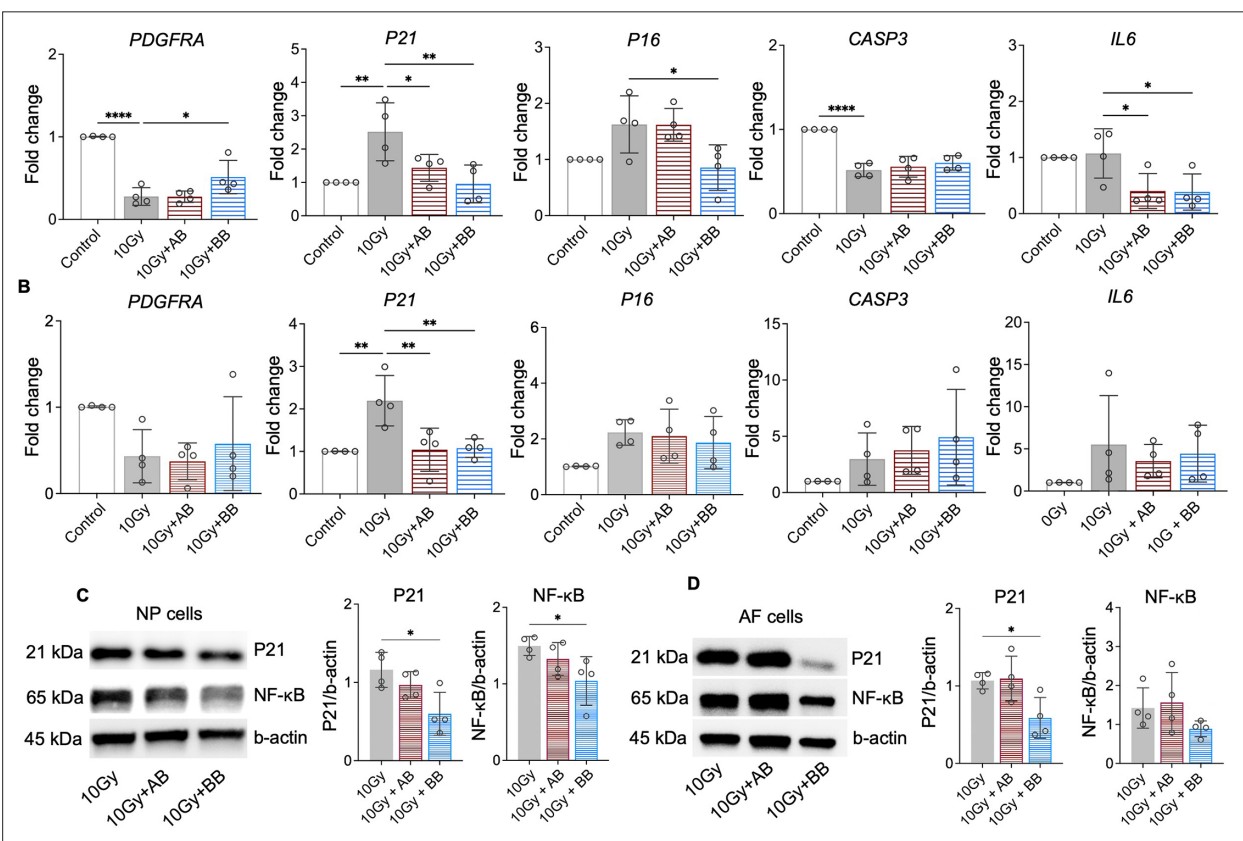

**Figure 6.** Platelet-derived growth factor (PDGF)-AB/BB treatment suppressed the expression of senescence-associated regulators. (**A, B**) Changes in the gene expression of *PDGFRA, P21, P16, CASP3,* and *IL6* in irradiated nucleus pulposus (NP) (**A**) and annulus fibrosus (AF) (**B**) cells treated with rhPDGF-AB/BB. (**C, D**) Protein expression levels of P21 and NF-κB in irradiated NP (**C**) and AF (**D**) cells treated with rhPDGF-AB/BB. n=4 each group. One-way ANOVA with Dunnett post hoc testing was performed. The data is presented as mean with SD. *p<0.05. **p<0.01. ****p<0.0001.

The online version of this article includes the following source data for figure 6:

**Source data 1.** Excel file containing the measures displayed in *Figure 6*.

**Source data 2.** PDF file containing the original western blots for *Figure 6*, indicating the bands and treatments.

**Source data 3.** Original files for western blots displayed in *Figure 6*.

Furthermore, compared to irradiation group, the protein levels of P21 and NF-κB, a major regulator stimulating SASP, were inhibited in rhPDGF-BB-treated NP cells (*Figure 6C*). In irradiated AF cells, the protein expression of P21 was significantly reduced by rhPDGF-BB treatment. The expression of NF-kB showed a decreasing trend in rhPDGF-BB-treated cells. rhPDGF-AB had no effects on the protein expression of P21 and NF-kB. Overall, PDGF-BB mitigated the progression of senescent phenotype in both human NP and AF cells.

## Discussion

The current study has shown for the first time that PDGF-AB and -BB treatment not only alleviated the senescent phenotype, but also suppressed the progression of cellular senescence in the IVD. Specifically, in aged, degenerated human NP and AF cells, PDGF-AB/BB treatment suppressed the ROS accumulation, mitochondrial dysfunction, and the resultant oxidative stress, while stimulating cell cycle progression. In the NP, PDGFRA was among the top hub genes that showed strong interaction with other genes. Moreover, its expression was significantly reduced in irradiation-induced senescent cells. Importantly, immediate PDGF-BB treatment after irradiation promoted the expression of *PDGFRA* and mitigated senescence-associated phenotype in healthy IVD cells. This novel function of PDGF-BB/PDGFRA signaling is likely linked to their ability to inhibit NF-κB, P16, and P21 pathways in NP cells. In contrast, while PDGF treatment alleviated the senescent phenotype in AF cells, it also induced changes in pathways such as response to mechanical stimuli and neurogenesis, which were distinct from those in NP cells. This indicates that the treatment enhanced IVD functionality through different mechanisms within the two compartments. Senescence is a key contributor to the onset and the progression of IVD degeneration (*Silwal et al., 2023*). It can occur naturally with aging or can be triggered by growth factor deficiency, radiation, DNA damage, oxidative stress, and inflammatory stimuli (*Wang et al., 2016*; *Huang et al., 2022*). Senescent cells exit the cell cycle and cannot divide; their accumulation reduces the IVD's ability to generate new cells to replace apoptotic or necrotic cells. It has been well established that an increased number of senescent cells was positively correlated with IVD aging and the severity of degeneration while negatively correlated with the percentage of proliferating cells in human NP and AF tissues (*Gruber et al., 2009*; *Veroutis et al., 2021*; *Gruber et al., 2007*). Since PDGF is a mitogenic growth factor and promotes cell proliferation in various cell types (*Contreras et al., 2021*; *García-Olivas et al., 2007*; *Martin-Garrido et al., 2013*), it is intriguing to explore whether PDGF-AB/BB could mitigate or prevent against cellular senescence in IVDs. We observed groups of upregulated genes regulating cell cycle progression and cell proliferation in PDGF-AB/BB-treated NP cells. AF cells also had an upregulation of chromosome segregation and mitotic nuclear division pathways, suggesting that these cells were actively proliferating. Cell cycle progression is controlled by cyclins, and the expression of cyclin D1 and B1, regulators of the G1/S and G2/M phase transition, respectively, was increased in degenerated NP cells after PDGF-AB/BB treatment. Transcriptomic analysis also revealed that increased cyclin D1 was among the top hub nodes, tightly regulated by PDGF-AB and BB. In addition, cell cycle analysis confirmed that PDGF-AB/BB induced the transition from G1 to S1 phase in senescent IVD cells. Our data suggested that PDGF treatment led to an increase in the proliferation of senescent IVD cells in the S phase likely through cyclin D1 and B1. These findings were in accordance with previous studies showing that PDGF-stimulated cell proliferation and cyclin D1 expression in rat mesangial cells (*Hida et al., 2003*), rat adventitial fibroblasts (*Chen et al., 2024*), and human vascular smooth muscle cells (*Martin-Garrido et al., 2013*).

The antisenescence of PDGF has been demonstrated in several studies. For example, PDGF-BB decreased SA-β-gal positive cells and the expression of P53 and P21 while enhancing proliferative potential in senescent human dermal fibroblasts (*Bae et al., 2016*) and mesenchymal stem cells (MSCs) derived from immune thrombocytopenia patients (*Zhang et al., 2016*). PDGF exerts its function by binding to its receptors, thereby triggering a cascade of intracellular signaling events, such as cell proliferation and survival. The PDGF-PDGFRA signaling is often suppressed in premature aged or senescent cells (*Tan et al., 2018*; *Pan et al., 2019*; *Mori et al., 1993*), suggesting that PDGF signaling dysfunction might be involved in senescence or aging. Notably, our transcriptome data has shown that both PDGF-AB and -BB induced the expression of *PDGFRA* in aged, degenerated NP cells. This autoregulatory loop caused IVD cells to become more sensitive to PDGF treatment and augmented the signaling cascades triggered by PDGF binding, including senescence-associated P21, P16, P53,

and NF-κB pathways. In line with this thought, PPI analysis revealed that *PDGFRA* closely interacted with upregulated *MDM2*, a negative regulator of P53. NF-κB signaling is a major signaling pathway that is involved in inflamm-aging (**Songkiatisak et al., 2022**) and induces the onset of SASP (**Salminen et al., 2012**), such as IL6, IL8, and CXCL1. In our study, irradiation in NP cells led to decreased *PDGFRA* expression, which impaired the activity of PDGF/PDGFRA signaling. However, PDGF-BB treatment promoted the expression of *PDGFRA* and decreased the expression of *P21, P16,* and *IL6* in senescent NP cells, likely by inhibiting NF-κB signaling. In fact, it has been shown that NF-kB signaling was elevated in mouse IVDs exposed to a single 20 Gy dose of irradiation in an ex vivo culture model (**Liu et al., 2020**). These findings support the idea that PDGF treatment could stimulate the expression of *PDGFRA* and mitigated the progression of senescent phenotype in healthy IVD cells. Interestingly, while mRNA level was increased in PDGF-treated NP cells, its protein level was decreased, highlighting the complexity in PDGF receptor dynamics. Upon binding with PDGF ligands, PDGFRA is known to undergo rapid internalization and degradation, a mechanism that prevents overstimulation of the signaling pathway (**Rogers and Fantauzzo, 2020**). The upregulated gene expression probably compensates for this degradation and supports continued activation of PDGFRA signaling activation, emphasizing its crucial role in response to the PDGF treatment. Further studies using preclinical animal models to validate the protectiveness of PDGF against the onset or progression of senescence in the IVD will offer more robust evidence.

Degenerated IVD cells experienced an imbalance between ROS accumulation and antioxidant capacity, leading to oxidative stress, which is involved in cellular senescence (**Feng et al., 2017**). Mitochondrion is the primary location of intracellular ROS production and its dysfunction in degenerated IVD cells causes electron leakage, resulting in the formation of superoxide radicals (**Wang et al., 2022**). PDGF-AB/BB treatment suppressed the activity of damaged mitochondria, leading to decreased ROS and oxidative stress in degenerated NP and AF cells, as indicated by GO BP analysis. Consistently, GSEA revealed the underrepresentation of oxidative phosphorylation in treated NP cells. The downregulation of genes related to mitochondria activity and oxidative phosphorylation may be indicative of a shift in metabolic pathways, leading cells to potentially prioritize alternative pathways such as glycolysis. Indeed, PDGF treatment upregulated groups of genes involved in response to decreased oxygen levels, suggesting that the treatment-activated cellular mechanisms involved in sensing and adapting to low oxygen conditions. In line with this idea, it has been demonstrated that hypoxia-inducible factor 1α is highly expressed in healthy NP cells to suppress oxidative phosphorylation and enhance glycolysis, allowing cells to adjust to a hypoxic microenvironment (**Song et al., 2024**; **Silagi et al., 2021**). Conversely, degenerated NP cells show an opposite metabolic pattern (**Song et al., 2024**; **Silagi et al., 2021**).

Besides the anti-senescence, NP and AF cells responded differently to PDGF-AB/BB treatment. AF cells were more responsive to the treatment with twice as many DEGs as NP cells. The difference in tissue architecture and cellular contents between NP and AF contributes to their specialized functions (**Kudelko et al., 2021**). The multiple layers of concentric lamellae in the AF enable it to withstand circumferential loads, while this highly organized structure was disrupted with aging and degeneration (**Dittmar et al., 2016**). In addition, damage to the AF has the potential to stimulate neovascularization and nerve ingrowth, provoking an inflammatory response and contributing to low back pain. PDGF-AB/BB treatment downregulated groups of genes related to neurogenesis and response to mechanical stimulus in AF cells, while downregulated genes in the NP were more associated with metabolic pathways, providing evidence that the treatment enhanced IVD functionality in different aspects between the two compartments.

Senescence as a central hallmark of aging can be induced by several factors (**Silwal et al., 2023**). One limitation of this study is that we did not examine the effects of PDGF-AB/BB treatment on senescence using other senescence models. In addition, our findings were obtained in vitro experiments, so caution is needed when interpreting our data. Our future work will focus on establishing an appropriate experimental animal model to study the functions of PDGF on cellular senescence.

In this study, we demonstrated a novel function of PDGF in human NP and AF cells. PDGF-AB/BB treatment alleviated oxidative stress and mitochondria dysfunction while stimulating cell cycle progression in degenerated IVD cells. In addition, the treatment mitigated the progression of senescence in healthy NP and AF cells following irradiation. These findings suggest that PDGF-AB/BB may serve as a promising potential candidate for IVD degeneration by targeting cellular senescence.

## Acknowledgements

The authors thank Gilbert Gu for his assistance in correcting the grammar in the manuscript. Funded by NIH-R01AR076427 to HD, and the Department of Orthopaedics at Emory University School of Medicine. The authors also express their gratitude to Mr. Jack McKee for his gift to support our departmental spine research.

## Additional information

### Funding

| Funder | Grant reference number | Author |
| --- | --- | --- |
| National Institute of Arthritis and Musculoskeletal and Skin Diseases | R01AR076427 | Hicham Drissi |
| Jack McKee | | Hicham Drissi |
| Emory University | Startup Funds | Hicham Drissi |

The funders had no role in study design, data collection and interpretation, or the decision to submit the work for publication.

### Author contributions

Changli Zhang, Conceptualization, Data curation, Software, Formal analysis, Supervision, Validation, Investigation, Visualization, Methodology, Writing – original draft, Writing – review and editing; Martha Elena Diaz-Hernandez, Conceptualization, Methodology, Writing – original draft, Writing – review and editing; Takanori Fukunaga, Sreekala Shenoy, Sangwook Tim Yoon, Methodology, Writing – review and editing; Lisbet Haglund, Conceptualization, Methodology, Writing – review and editing; Hicham Drissi, Conceptualization, Data curation, Supervision, Funding acquisition, Visualization, Writing – original draft, Project administration, Writing – review and editing

### Author ORCIDs

Changli Zhang ⓘ https://orcid.org/0000-0002-1170-0374
Sangwook Tim Yoon ⓘ https://orcid.org/0000-0003-1010-6952
Lisbet Haglund ⓘ https://orcid.org/0000-0002-1288-2149
Hicham Drissi ⓘ https://orcid.org/0000-0002-3322-281X

### Ethics

Human subjects: This study was conducted in accordance with the ethical principles outlined in the Declaration of Helsinki. All experiments and procedures were reviewed and approved by the institutional review board of Emory University (IRB #00099028). Tissues were obtained with written informed consent from each donor.

Reviewer #1 (Public review): https://doi.org/10.7554/eLife.103073.3.sa1
Reviewer #2 (Public review): https://doi.org/10.7554/eLife.103073.3.sa2
Author response https://doi.org/10.7554/eLife.103073.3.sa3

## Additional files

### Supplementary files

MDAR checklist

### Data availability

Sequencing data have been deposited in NCBI Sequence Read Archive database under the accession code PRJNA1150962. All data generated or analysed during this study are included in the manuscript and supporting files.

The following dataset was generated:

| Author(s) | Year | Dataset title | Dataset URL | Database and Identifier |
|-----------|------|---------------|-------------|-------------------------|
| Zhang C, Elena Diaz-Hernandez M, Fukunaga T, Sreekala S, Yoon ST, Haglund L, Drissi H | 2024 | Anti-senescence of PDGF in human intervertebral disc | https://www.ncbi.nlm.nih.gov/bioproject/PRJNA1150962 | NCBI BioProject, PRJNA1150962 |

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
