## [Editor Report · eLife Assessment]

This work demonstrates the therapeutic potential of recombinant human PDGF-AB/BB proteins in alleviating the senescent signatures of primary human intervertebral disc cells. The study represents a **fundamental**, significant advance in the treatment of intervertebral disc degeneration through the suppression of senescence. The strength of evidence supporting these conclusions is **compelling**, as it is primarily based on transcriptomic analysis and direct protein measurements from relatively homogeneous cell populations. This work will be of interest to spine basic scientists and clinicians, as well as to musculoskeletal scientists more broadly. The revised manuscript adds greater clarity, and the impact of the study is greatly enhanced.

---

## [Referee Report · Reviewer #1 (Public review)]

The authors, Zhang et al., demonstrate the beneficial effects of treating degenerate human primary intervertebral disc (IVD) cells with recombinant human PDGF-AB/BB on the senescence transcriptomic signatures. Utilizing a combination of degenerate cells from elderly humans and experimentally induced senescence in young, healthy IVD cells, the authors show the therapeutic effects on mRNA transcription as well as cellular processes through informatics approaches.

One notable strength of this study is the use of human primary cells and recombinant forms of human PDGF-AB/BB proteins, which increases the translational potential of these in vitro studies. The manuscript is well-written, and the informatics analyses are thorough and clearly presented.

Comments on revisions:

The revised manuscript adds greater clarity, and the impact of the study is greatly enhanced.

---

## [Referee Report · Reviewer #2 (Public review)]

Summary:

This work highlights a novel role for platelet-derived growth factor (PDGF) in mitigating cellular senescence associated with age-related and painful intervertebral disc degeneration. Prior literature has demonstrated the importance of accumulation of senescent cells in mediating many of the pathological effects associated with the degenerate disc joint, such as inflammation and tissue breakdown. In this study, the authors treat clinically relevant human nucleus pulposus and annulus fibrosus cells from patients undergoing discectomy with recombinant PDGF-AB/BB for 5 days and then deep phenotyped the outcomes using bulk RNA sequencing. In addition they irradiated healthy human disc cells which they subsequently treated with PDGF-AB/BB examining the expression of SASP-related markers and also PDGFRA receptor gene expression. Overall PDGF was able to down-regulate many senescent associated pathways and the degenerate phenotype in IVD cells. Altered pathways were associated with neurogenesis, mechanical stimuli, metabolism, cell cycle, reactive oxygen species and mitochondrial dysfunction. Overall the authors achieved their aims and the results by and large support their conclusions although improvements could be made to enhance the rigor of the study and findings

Strengths:

A major strength of this study is the use of human cells from patients undergoing discectomy for disc herniation as well as access to healthy human cells. Investigating the role of PDGF regarding cellular senescence in the degenerate disc joint is novel and an underexplored area of research which is a significant contribution to the field of spine. This study highlights a potential target for addressing cellular senescence where most of the prior focus has been on senolytic drugs. Such studies have broad implications to other age-related diseases where senescence plays a major role. The use of transcriptomics and therefore an unbiased approach to investigating the role of PDGF is also considered a strength as is the follow-up studies involving irradiating healthy human disc cells and treating these cells with PDGF. The combined assessment of both nucleus pulposus and annulus fibrosus cells in the context of these studies adds to the impact.

Weaknesses:

A weakness of these studies relates to qualitative data presented for the B-galactosidase assay. Quantification of such data sets would greatly strengthen the studies and lend further support to the hypotheses. The study in its current form could be strengthened by the inclusion of mechanistic studies probing the downstream PDGF receptor associated pathways for example specifically targeting or modulating the activity of the PDGF receptor PDGFRA.

---

## [Author Response]

The following is the authors’ response to the original reviews

**Reviewer #1:**
The Reviewer asks that we provide the source of PDGF-AB/BB proteins.

We apologize for omitting such information. We now provide the source of PDGF-AB/BB in the Methods as PeproTech. In our revised manuscript we clearly state in Page 7, line 142: “Cells were then treated with recombinant human PDGF-AB (40ng/ml; PeproTech, 10770584) or -BB (20ng/ml; PeproTech, 10771918) for 5 days. “

The Reviewer asks that we adequately report our chosen irradiation parameters suggesting that we consider (PMCID: PMC5495460) for appropriate parameter reporting.

We thank the Reviewer for this excellent suggestion. We now provide a more detailed irradiation reporting based on the shared manuscript in Page 9, line 10, line 204.

The Reviewer requests more details about the age range to distinguish young from old donors.

In the Methods section of our revised manuscript, we now provide the age range for our old donors being between 53 and 67 while our younger donor population ranged between 19 and 27 years of age. These changes are reflected in Page 6, line 128: “Human degenerated NP and AF tissues (Grade IV or V on Pfirrman grade; 64.6 ±8.5 years old) were obtained as the surgical waste from donors with discogenic pain, with each donor providing written informed consent. Healthy NP and AF cells (23.0 ±3.7 years old) were gifted by Professor Lisbet Haglund from McGill University (Tissue Biobank #2019-4896).”

The Reviewer wonders about the rationale for using different concentrations of PDGF-AB/BB in the degenerate cell and irradiation experiments.

We apologize for our lack of clarity. We initially treated cells with different concentrations (20 and 40 ng/ml) of PDGF-AB/BB to first establish a dose-response. From our MTT and gene expression analyses we determined that 20ng/ml was sufficient to elicit significant changes in cell proliferation markers, including MKI67, CCNB1 and CCND1. Increasing the concentration to 40 ng/ml of either growth factor did not significantly influence these parameters. However, we felt that for our bulk RNA seq experiments, we may see better changes in signaling molecules under 40ng/ml of PDGF-AB since its effects on cell growth at this concentration were maximal while PDGF-BB was maintained at 20ng/ml based on its efficacy in our mitogenic response.

The Reviewer asks that we consider describing the effects of PDGF-AB/BB as mitigating or therapeutic rather than protective both in the title and throughout the manuscript.

We agree with the Reviewer’s recommendation, and we have now changed the title to “Therapeutic effects of PDGF-AB/BB against cellular senescence in human intervertebral disc”. Moreover, we implemented this change in the revised manuscript as requested.

The Reviewer believes that changes in the NP are more clinically evident (by imaging methods), despite degeneration often initiating from the AF (annulus fibrosus), e,g. through tears/microtears and would like for us to reflect this in our revised manuscript.

We agree with the Reviewer’s comment, and we thank them for this added accuracy. On this basis, we now corrected our language in the introduction by stating in Page 4, line 68 that: “To date, the main focus of IVD cell studies has been on the NP, as changes in the NP are easily detected through imaging techniques like MRI, making it the most visible indicator of disc degeneration in clinical practice. In addition, NP plays a crucial role in the progression of IVD degeneration due to its susceptibility to significant structural and functional changes during aging and degeneration.”

The Reviewer points out a prior study which examined the effects of X-ray irradiation on NF-kB signaling in young and aged IVDs (PMCID: PMC5495460) suggesting that we include this reference in our revised manuscript.

We thank the Reviewer for this suggestion, and we are now referencing this elegant study in the discussion section of our revised manuscript. Thus, in page 20, line 440 we state: “ In fact, it has been shown that NF-kB signaling was elevated in mouse IVDs exposed to a single 20 Gy dose of irradiation in an ex vivo culture model.”

The Reviewer asks that our experimental methods are described in the order of the experimental workflow. For example, section 2.2 describes RNA sequencing, which is a terminal assay. Section 2.2 may be more appropriate for detailing the methods of PDGF-AB/BB treatment, along with the rationale.

We thank the Reviewer for pointing this out and have reorganized the Methods section accordingly.

**Reviewer #2:**
The Reviewer requests more experimental details in the methodology including the rationale for such methods/conditions as well as specific culture models utilized, substrates, cell density, and media components.

We apologize for our lack of clarity. We now revised the methods section based on the comments.

The Reviewer asks about the quantitative data for b-galactosidase assay and immunofluorescence of senescence-associated proteins such as P21 and P16.

We apologize for omitting this information. We now included the quantification of P21 and P16 positive cells, which is presented in the revised Figures 4. For b-galactosidase assay, we were unable to quantify the percentage of positive cells because we did not perform nuclei staining, making it difficult to accurately determine the total cell number. Instead, we provided representative images showing the full field of view at 10X magnification using Echo microscope.

The Reviewer requests the protein level data of PDGFRA to determine if the transcripts are being translated to protein.

We thank the Reviewer for this suggestion. The protein expression of PDGFRA has been included in the Supplementary Figure 2. We found that PDGFRA protein levels were decreased in both NP and AF cells in response to PDGF treatments. It is known that upon binding with PDGF ligands, PDGFRA undergoes rapid internalization and degradation, a mechanism that prevents overstimulation of the signaling pathway (doi: 10.1042/BST20200004). The upregulated gene expression probably attempting to compensate for this degradation and supports continued activation of PDGFRA signaling activation, emphasizing its crucial role in response to the PDGF treatment. Thus, we implemented it in the discussion section in page 22, line486:” Interestingly, while mRNA level was increased in PDGF treated NP cells, its protein level was decreased, highlighting the complexity in PDGF receptor dynamics. Upon binding with PDGF ligands, PDGFRA is known to undergo rapid internalization and degradation, a mechanism that prevents overstimulation of the signaling pathway (Rogers and Fantauzzo 2020). The upregulated gene expression probably attempting to compensate for this degradation and supports continued activation of PDGFRA signaling activation, emphasizing its crucial role in response to the PDGF treatment.”

The Reviewer points out that our conclusion that “PDGF do not mediate their effects via the PDGFRA” is not supported by the current data asking that further discussion, interpretation, and direct comparison of the nucleus pulposus and annulus fibrosus data sets be presented to the readers.

We thank the Reviewer for the insightful comment. In page 20, line 432, we have corrected our language to now state: “In contrast, while PDGF treatment alleviated the senescent phenotype in AF cells, it also induced changes in pathways such as response to mechanical stimuli and neurogenesis, which were distinct from those in NP cells. This indicates that the treatment enhanced IVD functionality through different mechanisms within the two compartments.”

The Reviewer cannot appreciate the changes in S-phase between control and treated groups.

We apologize for the poor quality of the figure in our initial submission. We analyzed the data in S phase and included them in our revised Figures 5C and 5F.

The Reviewer believes that discectomies are typically not performed on patients with discogenic back pain but on patients who are undergoing surgery for a herniated disc.

We agree with the Reviewer, and we corrected our language in the revised manuscript. In Page 6, line 128, we now stated: “Human degenerated NP and AF tissues (Grade IV or V on Pfirrman grade; 64.6 ±8.5 years old) were obtained as the surgical waste from donors with disc herniation, with each donor providing written informed consent.”

The Reviewer asks about the protein-protein interactions in AF cells.

We thank the Reviewer for this suggestion, and we now included it in Figure 3.

The Reviewer requests more details about the protocol and doses for the irradiation studies.

In the revised manuscript, we added this information in page 10, line 204.

The Reviewer asks whether the gene expression of PDGFRA was increased or decreased in irradiated cells compared to non-irradiated cells.

The gene expression of PDGFRA was decreased in NP cells exposed to irradiation compared to non-irradiated cells. The data are shown in Figure 4 and their description in the text is in page 17, line 411.